

# *Symbiodinium* spp. associated with scleractinian corals from Dongsha Atoll (Pratas), Taiwan, in the South China Sea

Shashank Keshavmurthy[1], Kuo-Hsun Tang[1,2], Chia-Min Hsu[1], Chai-Hsia Gan[1], Chao-Yang Kuo[1,3], Keryea Soong[4], Hong-Nong Chou[2] and Chaolun Allen Chen[1,5,6]

[1] Biodiversity Research Center, Academia Sinica, Taipei, Taiwan
[2] Institute of Fisheries Science, National Taiwan University, Taipei, Taiwan
[3] ARC Centre of Excellence for Coral Reef Studies, James Cook University, Townsville, Queensland, Australia
[4] Department of Oceanography, Dongsha Atoll Research Station, National Sun Yat-Sen University, Kaohsuing, Taiwan
[5] Taiwan International Graduate Program (TIGP)-Biodiversity, Academia Sinica, Taipei, Taiwan
[6] Institute of Oceanography, National Taiwan University, Taipei, Taiwan

Corresponding authors
Shashank Keshavmurthy,
shashank@gate.sinica.edu.tw,
coralresearchtaiwan@gmail.com
Chaolun
Allen Chen, cac@gate.sinica.edu.tw

## ABSTRACT

Dongsha Atoll (also known as Pratas) in Taiwan is the northernmost atoll in the South China Sea and a designated marine national park since 2007. The marine park's scope of protection covers the bio-resources of its waters in addition to uplands, so it is important to have data logging information and analyses of marine flora and fauna, including their physiology, ecology, and genetics. As part of this effort, we investigated *Symbiodinium* associations in scleractinian corals from Dongsha Atoll through surveys carried out at two depth ranges (shallow, 1–5 m; and deep, 10–15 m) in 2009 and during a bleaching event in 2010. *Symbiodinium* composition was assessed using restriction fragment length polymorphism (RFLP) of 28S nuclear large subunit ribosomal DNA (nlsrDNA). Our results showed that the 796 coral samples from seven families and 20 genera collected in 2009 and 132 coral samples from seven families and 12 genera collected in 2010 were associated with *Symbiodinium* C, D and C+D. Occurrence of clade D in shallow water (24.5%) was higher compared to deep (14.9%). Due to a bleaching event in 2010, up to 80% of coral species associated with *Symbiodinium* C underwent moderate to severe bleaching. Using the fine resolution technique of denaturing gradient gel electrophoresis (DGGE) of internal transcribed spacer 2 (ITS2) in 175 randomly selected coral samples, from 2009 and 2010, eight *Symbiodinium* C types and two *Symbiodinium* D types were detected. This study is the first baseline survey on *Symbiodinium* associations in the corals of Dongsha Atoll in the South China Sea, and it shows the dominance of *Symbiodinium* clade C in the population.

## INTRODUCTION

Coral reefs provide habitat for numerous marine organisms and are considered among the richest ecosystems on earth. In addition to their ecological values, they are also economically

important as they contribute toward food, tourism, coastal protection, aesthetics, and cultural significance to people in coastal areas (*Moberg & Folke, 1999*; *Hoegh-Guldberg, 2004*; *Wilkinson, 2004*) and act as ecosystem engineers (*Jones, Lawton & Shachak, 1994*; *Coleman & Williams, 2002*). As a result of climate change and anthropogenic disturbances, corals and coral reefs in the recent decades (30–40 years) have suffered an unprecedented decline in terms of species abundance and community degradation (*Hoegh-Guldberg, 1999*; *Coles & Brown, 2003*; *Hughes et al., 2003*; *Bellwood et al., 2004*).

Increasing seawater temperature is considered one of the main causes of this decline (*Hoegh-Guldberg et al., 2007*). Generally, corals undergo a phenomenon known as bleaching when confronted with 1.0–2.0 °C above the mean summer average seawater temperatures. Bleaching is a result of the breakdown of symbiosis between the coral host and single celled algae *Symbiodinium*, either due to release of *Symbiodinium* cells by the host or escape of the cells from the host (see *Weis, 2008*). While a majority of corals undergo bleaching, some coral species can resist thermal stress or changes in environmental conditions. This is due either to the ability of the coral host to withstand stress or by associating with a more thermally tolerant type of *Symbiodinium*, or to a combination of the two (*Berkelmans & Van Oppen, 2006*; *Bhagooli, Baird & Ralph, 2008*; *Baird et al., 2009*).

Based on various genetic markers (*Rowan & Powers, 1991a*; *Rowan & Powers, 1991b*; *LaJeunesse, 2001*; *Van Oppen et al., 2001*; *Pochon et al., 2001*; *Fabricius et al., 2004*; *Coffroth & Santos, 2005*; *Pochon et al., 2006*; *Stat et al., 2009*), *Symbiodinium* are genetically diverse and has been classified into clades, types and ecomorphs. Different host–symbiont assemblages can respond differently to diverse conditions, such as temperature, irradiance, and sedimentation disturbance. Most *Symbiodinium* clades/types are associated with specific host genera or species (*LaJeunesse et al., 2004*). In many cases, *Symbiodinium* clades/types associate with only one or a few host species. However, studies also have shown that one of the two partners is more flexible (*Baker, 2003a*) and occupies defined ecological niches and roles within and across coral hosts (*LaJeunesse et al., 2010b*; *Pochon & Gates, 2010*; *Weber & Medina, 2012*) based on their physiological responses to various environmental stresses (*Iglesias-Prieto et al., 2004*; *Baker, 2003a*; *Baker, 2003b*; *Jones et al., 2008*; *Little, Van Oppen & Willis, 2004*; *Rowan, 2004*; *Sampayo et al., 2008*; *Warner et al., 2006*).

*Symbiodinium*-related stress-resistant mechanisms could be a result of strict association with a stress resistant clade/type of *Symbiodinium* or, in case of coral host with multiple *Symbiodinium* association, the result of shuffling between stress sensitive and stress resistant *Symbiodinium* clades/types. The capacity to shuffle *Symbiodinium* clades/types may be key for their acclimatization and/or adaptation (*Baker & Romanski, 2007*; *Mieog et al., 2007*). The general trend is for thermal-tolerant clades/types to supersede thermal-sensitive clades/types under temperature stress (*Jones et al., 2008*; *Jones & Berkelmans, 2010*).

As coral and coral reefs face more frequent and intense bleaching events due to climate change and anthropogenic stressors, it is necessary to document the coral-*Symbiodinium* associations from locations which have been ignored or those that are remote. A recent study from western Australia have shown presence of some unique *Symbiodinium* types as well as added to the information on coral-*Symbiodinium* associations and *Symbiodinium*

diversity (*Silverstein et al., 2011*). Such studies will help us to understand the current status and predict future response of the corals in new and remote locations.

Dongsha Atoll in the South China Sea is one such remote location. Although many studies have been conducted on coral-*Symbiodinium* associations in the South China Sea, no such efforts have been realized in Dongsha Atoll. The Dongsha Atoll Marine National Park was established in 2007 in order to implement legal jurisdiction for conservation efforts. Several studies have been conducted on its reef-building corals despite its remote location. For example, *Ma (1937)* studied growth rates in different coral species. Coral biodiversity (17 genera and 45 species) was first described by *Yang et al. (1975)*, followed by *Fang et al. (1990)*, who reported 28 genera and 63 species. Later, *Dai, Fan & Wu (1995)* recorded 34 genera and 101 species of scleractinian corals, eight genera and 33 species of octocorals, and one genus and three species of hydrocorallina. Recently, and especially after 1998, interest has arisen on the effects of mass coral bleaching of the coral reef community of Dongsha Atoll. After the exceptionally high 1997–98 summer temperatures during an *El Niño–Southern Oscillation* (ENSO) bleaching event, different studies reported a decrease in its coral cover and biodiversity (*Fang, 1998*; *Li & Fang, 2002*; *Soong, Dai & Lee, 2002*). Moreover, *Li et al. (2000)* showed a decline in coral species number due to the extensive use of poisons and explosives from illegal fishing. Before 1998, *Porites* and *Acropora* were the most abundant and widespread genera in Dongsha (*Fang et al., 1990*; *Dai, Fan & Wu, 1995*). Although *Porites* has remained a dominant genus from 1998 to the present, it has shared its dominance with other species belonging to the Merulinidae and Fungiidae (*Dai, Qin & Zheng, 2013*).

Although the species diversity of scleractinian corals at Dongsha has been investigated extensively (*Fang et al., 1990*; *Li et al., 2000*; *Soong, Dai & Lee, 2002*; *Dai, Fan & Wu, 1995*; *Dai, 2005*; *Dai, 2006*; *Dai, 2008*; *Dai, Qin & Zheng, 2013*), no studies have been carried out on the coral-*Symbiodinium* associations. Dongsha Atoll is characterized by well-developed tropical atoll reefs with 281 known scleractinian coral species (*Dai, Qin & Zheng, 2013*). Direct anthropogenic disturbances on the atoll are minimal (about 200 military personnel and park rangers) since Dongsha is a military-exclusive area and a designated marine national park. However, there is occasional human disturbance by fishermen from nearby coastal nations, including China, Philippines, and Vietnam. Hence, the aim of this study was to conduct a baseline survey on *Symbiodinium* associations in the corals of Dongsha Atoll and document the *Symbiodinium* diversity in shallow and deep waters at different sites in the lagoon of the Atoll in 2009 and 2010. In this paper, we report the results from the preliminary survey carried out in the lagoon of Dongsha Atoll on coral-*Symbiodinium* associations.

## MATERIALS AND METHODS
### Description of the study site and sample collection
Dongsha Atoll (20°35′–47′N and 116°41′–55′E), also known as Pratas Island (Fig. 1), is located in the northern South China Sea, 850 km southwest of Taiwan. Dongsha is a circular atoll about 25 km in diameter and its central lagoon covers more than 600 km$^2$.

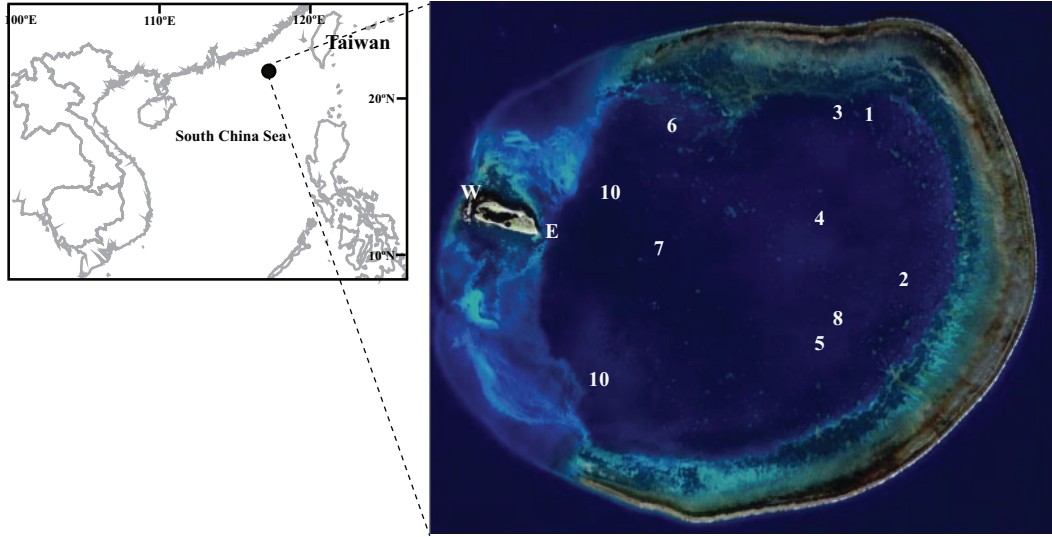

**Figure 1** **Figure of sampling location.** Map of Dongsha Atoll in the South China Sea. Sampling sites in the lagoon are marked 1–10.

Depths in the lagoon range 10–15 m with a maximum depth of 23.7 m (*Dai, Qin & Zheng, 2013*). Composed of coral debris, Dongsha Island is small (2,860 m long 865 m wide), 2 m above sea level at most, covers a total surface area of 1.74 km$^2$, and is located in the western part of the atoll. The circular reef flat around the lagoon spreads over 46 km and its maximum width reaches 2 km.

Coral samples were collected in June and September 2009 and September 2010 in the Dongsha Atoll Lagoon (DAL) (Fig. 1). There was a bleaching episode during summer of 2010 due to elevated seawater temperatures. The corals underwent bleaching mostly in the shallow waters of DAL. Bleached and non-bleached corals were determined by visual survey of color variation between healthy and bleached corals. Sampling was limited to the lagoon due to military-restricted access to the reef crest and outside the atoll. A total of 928 scleractinian coral samples including seven families with 20 and 12 genera in 2009 and 2010 respectively, were collected at two depths: 1–5 m (shallow) and 10–15 m (deep). Samples in 2009 and 2010 were collected from nine and four sites, respectively. Samples (∼5–10 cm) form coral colonies were randomly collected by hammer and chisel. All the samples were preserved in 70% ethanol (v/v) and kept at 4 °C until processing. Scleractinian corals were identified to species when possible. Scleractinian taxonomy followed *Veron & Stafford-Smith (2000)*, *Dai & Hung (2009a)*, *Dai & Hung (2009b)*, and *Huang et al. (2014)*. *In situ* seawater temperatures at every site and at both depths were recorded with temperature loggers (HOBO, Pendant$^{TM}$, Onset Computer Corporation, MA, USA) at 30 min intervals (the periods of record were June 17 to September 15, 2009, and May 28 to September 6, 2010).

## DNA extraction

Total genomic DNA (coral host + *Symbiodinium*) was extracted by the salting-out method (*Ferrara et al., 2006*). Coral tissue was lysed overnight in a 2 mL Eppendorf tube with 200

µL of lysis buffer (0.25 M Tris, 0.05 M EDTA at pH 8.0, 2% sodium dodecylsulfate (SDS), and 0.1 M NaCl) and 10 µL of 10 mg mL$^{-1}$ proteinase E at 55 °C in a water bath. NaCl (210 µL at 7 M) was added to the lysed tissue in the tube and the sample mixed carefully by inverting the tube. The solution was then transferred to a 2 mL collection tube containing a DNA spin column (Viogene, Sunnyvale, CA, USA) and centrifuged at 8,000 rpm for 1 min. The lysate was washed twice with 500 µL of ethanol (70%) by centrifuging at 8,000 rpm for 1 min at each step, with an additional centrifugation step at 8,000 rpm for 3 min to dry the spin column. The column was dried further at 37 °C for 15 min and the DNA then eluted by adding 150 µL of distilled water, with a final centrifugation at 14,000 g for 3 min. The quality of genomic DNA was checked using a 1% agarose gel and the concentration of eluted DNA was examined using NanoDrop and then stored at −20 °C for further analysis.

### Molecular identification of Symbiodinium clades (28S RFLP) and types (ITS2 DGGE)

A total of 928 samples (20 genera in 2009 and 12 genera in 2010) were used to analyze *Symbiodinium* clade composition using RFLP method modified from *Chen et al. (2005a)* and *Chen et al. (2005b)*. The 5′ end of nuclear large subunit ribosomal DNA (nlsrDNA) was amplified using a *Symbiodinium*-specific primer set, 28Szoox-D1/D2F (5′-CCT CAG TAA TGG CGA ATG AAC A-3′) and 28Szoox-D1/D2R (5′-CCT TGG TCC GTG TTT CAA GA-3′) (*Loh et al., 2001*). A 25 µl PCR reaction consisting of 3 µl DNA (10 ng µl$^{-1}$), 2 µl dNTPs (0.8 mM), 2 µl forward primer (0.16 µM), 2 µl reverse primer (0.16 µM), 3 µl PCR Buffer (1.2X), 0.5 unit *Taq* polymerase (Protech, Taipei, Taiwan), and 12.5 µl distilled water was run on a Px2 thermal cycler (Thermo Scientific, Waltham, MA, USA). The PCR cycling profile consisted of initial denaturation at 95 °C for 1 min followed by 5 cycles of 94 °C for 30 s, 30 cycles of annealing at 55 °C for 1 min and decreasing 1 °C to a final annealing temperatures of 50 °C and 2 min at 72 °C. The final extension was at 72 °C for 10 min. The PCR product was digested with Rsa I (BioLabs, San Diego, CA, USA) solution at 37 °C overnight and then run on 3% agarose gel (2% low melting agarose with 1% agarose) at 50 V for 3 h. Bands were stained using ethidium bromide (EtBr) and visualized under ultraviolet radiation.

For ITS2 *Symbiodinium* type composition, 168 samples were randomly picked from 17 genera in 2009 and 11 genera in 2010. The ribosomal internal transcribed spacer 2 (ITS2) region of *Symbiodinium* was amplified using the primers ITSintfor2, 5′-GAA TTG CAG AAC TCC GTG-3′, and ITS2clamp, 5′-CGC CCG CCG CGC CCC GCG CCC GTC CCG CGG GAT CCA TAT GCT AAG TTC AGCGG GT-3′ (*LaJeunesse & Trench, 2000*). Each 50 µl polymerase chain reaction (PCR) comprised 50 ng of genomic DNA, PCR buffer, 2.5 mM MgCl$_2$, 0.4 mM dNTPs, 0.4 µM of each primer, and 2 units of Taq polymerase (Invitrogen, Carlsbad, CA, USA). The PCR was run on a Px2 thermal cycler (Thermo Scientific, Waltham, MA, USA) with a touch-down PCR (*LaJeunesse, 2002*) to ensure specificity. An initial denaturation period at 92 °C for 3 min was followed by 20 cycles of 30 s at 92 °C, annealing from 62 °C to a final annealing temperature of 52 °C with decrements of 0.5 °C, and 30 s at 72 °C. Once the annealing temperature reached 52 °C, a further 20 cycles were performed at that annealing temperature, with a final extension period

of 10 min at 72 °C. Each PCR product along with *Symbiodinium* type markers (known ITS2 sequences confirmed by comparing with NCBI GenBank database) was loaded onto an acrylamide denaturing gradient gel (45–80%) and electrophoresed at 115 V for 15 h using a CBS Scientific system (Del Mar, CA, USA). Gels were stained with SYBR Green (Molecular Probes, Eugene, OR, USA) for 15 min and photographed for further analysis. Band patterns were confirmed by sequencing the bands cut from the DGGE gel.

## Statistical analysis

Shallow and deep-water temperature comparisons were analyzed by paired $t$-tests and average daily seawater temperature values presented as means $\pm$ standard deviation (SD). The frequency of daily maximum temperature fluctuation at each location and depth was calculated followed by difference in temperature between shallow and deep waters. Finally, the percentage of daily average seawater temperature of over 30 °C during the monitoring period was calculated. The *Symbiodinium* clade values in 2009 and 2010 were converted to percentage within and between genera and within and between two depths. The proportion of *Symbiodinium* clades in coral genera in 2009 and 2010 and between shallow and deep waters was analyzed using a chi-square test. All graphs were drawn using Aabel (Ver. 3.0; Gigawiz Ltd. Co., Tulsa, OK, USA) or Datagraph (Visual Data Tools) software for the Macintosh platform.

## RESULTS

### Seawater temperature

Seawater temperature was recorded (30 min intervals) from June 17 to September 15, 2009, and May 28 to September 6, 2010 (Figs. 2A and 2B, Data S1–Data S3). During the 2009 period, daily average temperatures at shallow waters (29.55 $\pm$ 0.23 °C) were higher than at deep waters (28.82 $\pm$ 0.19 °C; Fig. 2A; Table 1; paired $t$-test, $p < 0.001$). Similarly, in 2010, temperatures at shallow waters (29.94 $\pm$ 0.29 °C) were higher than at deep waters (29.72 $\pm$ 0.32 °C; Fig. 2A; Table 1; paired $t$-test, $p < 0.001$). Overall, $\Delta$T was 0.62 $\pm$ 0.3 °C in 2009 and 0.22 $\pm$ 0.14 °C in 2010. Daily temperature differences between shallow and deep waters in 2009 were 0.5–1.0 °C and 0–0.5 °C in 2010. In 2009, during the period monitored, 36% of the temperatures (number of days) recorded at the shallow waters exceeded 30.0 °C, whereas only 13% of the temperatures (number of days) recorded at the deep waters reached this value. In 2010, for shallow and deep waters, 61% and 55% of the temperatures (number of days) recorded exceeded 30.0 °C. This was also confirmed by the heat map of seawater temperatures measured at reef base and reef top during June–September and May–September in 2009 and 2010 (Fig. 2B).

In this study, samples were collected from nine locations in the DAL. Since the sampling was random and uneven within and between locations and between two sampling times (2009 and 2010), we present results from 2009 and 2010 by combining the data for locations and depths, rather than discussing the results from individual locations and depths. Hence, we present the overall picture of coral-*Symbiodinium* associations in DAL. Data of *Symbiodinium* clades in coral genera at different locations is provided as Data S4.

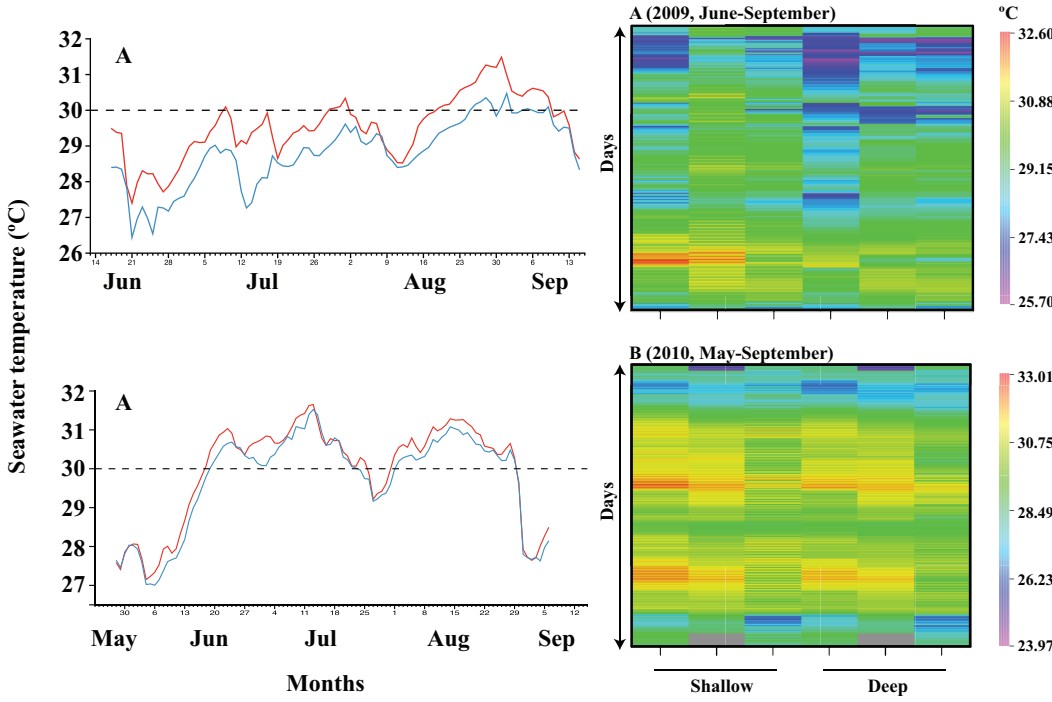

**Figure 2** **Seawater temperature at the study site in 2009 and 2010.** Seawater temperature in 2009: June–September (A); and 2010: May–September (B). The red line represents shallow and the blue line represents deep. Heat map of seawater temperatures measured at reef base and reef top during June–September and May–September in 2009 (A) and 2010 (B), respectively. The y-axis values are seawater temperature recorded at 30 min intervals.

**Table 1** **Percentage of daily average seawater temperature over 30 °C recorded during the sampling period in 2009 and 2010 within each location.**

| Temperature over 30 °C | 2009 | 2010 |
|---|---|---|
| Reef 1 Top | | 68.10% |
| Reef 1 Base | 7.30% | 62.30% |
| Reef 2 Base | 3.20% | |
| Reef 4 Top | | 65.50% |
| Reef 4 Base | | 64.30% |
| Reef 5 Top | 23.90% | |
| Reef 5 Base | 10.40% | |
| Reef 6 Top | 52.70% | |
| Reef 6 Base | 19.40% | |
| Reef 9 Top | 31.20% | 48.80% |
| Reef 9 Base | 23.60% | 39.80% |
| Reef Top over 30 °C | 35.90% | 60.80% |
| Reef Base over 30 °C | 12.80% | 55.50% |

**Notes.**
Top, Shallow; Base, Deep.

### *Symbiodinium* clade composition in 2009

A total of 796 samples from 20 genera and seven families were collected in 2009 (Figs. 3 and 4). RFLP band patterns revealed the presence of *Symbiodinium* clades C, D and C + D.

*Symbiodinium* Clade C occurred in all 20 genera at shallow and deep waters in 2009. *Leptastrea* (100%), *Fungia* (100%), and *Porites* (100%) were found to harbor only *Symbiodinium* clade C at shallow and deep waters. *Cyphastrea* (100%), *Hydnophora* (100%), *Montipora* (100%), and *Psammocora* (100%) were associated only with *Symbiodinium* clade C at deep waters. At shallow waters, the proportion of *Symbiodinium* clades was 64.18% (clade C), 24.5% (clade D), and 11.34% (clades C + D). And, at deep waters, the proportion of *Symbiodinium* clades was 78.6% (clade C), 14.9% (clade D), and 6.53% (clades C + D). The clade D proportion in comparison to clade C was significantly different at shallow and deep waters (chi-square test, $p = 0.007$). The clade D proportion was high at shallow waters (24.5%) compared to deep waters (14.9%). At both depths, most corals were associated with >50% of *Symbiodinium* clade C, the exceptions being *Oxypora* (33 % in shallow and 44% in deep), *Hydnophora* (22% in shallow), *Echinopora* (9% in shallow and 24% in deep), *Favites* (33% in shallow), *Coelastrea* (37% in shallow), *Astrea* (25% in shallow), *Pectinia* (38% in shallow) *and Goniopora* (43% in shallow), which were associated >50% of the time with clades D or C + D. However, care should be taken when interpreting the results from this data since the sample number was less than ten for most of these genera.

Compared to other genera, Clade D proportions in *Turbinaria* and *Coelastrea* were significantly different when comparing shallow to deep (chi-square test, $p < 0.05$). The clade D proportion of *Turbinaria* at shallow waters (39.5%) was higher than at deep waters (10.8%), and in *Coelastrea* was 46.7% at shallow waters and 15% at deep waters.

### *Symbiodinium* clade composition in corals during the 2010 bleaching episode

A total of 132 samples from 12 genera and seven families were sampled in 2010 from shallow and deep waters (Figs. 3 and 4). *Symbiodinium* clade D showed significant differences (chi-square test, $p < 0.001$) between bleached and non-bleached corals. Clade D proportions were extremely high in non-bleached corals (62.8%) compared to bleached corals (10.9%). Bleached corals were dominated by *Symbiodinium* clade C at each location. *Astreopora, Psammocora, Fungia, Cyphastrea, Porites,* and *Oxypora,* all associated with clade C, experienced bleaching in Dongsha Lagoon in the summer of 2010, and *Pavona, Coelastrea* and *Echinopora* still experienced bleaching even though they were associated with clades D or C + D.

### *Symbiodinium* type composition in corals in 2009 and 2010

There was no particular trend in the *Symbiodinium* type composition between the two sampling years due to random sampling of the coral colonies (Table 2). However, *Symbiodinium* composition was dominated by different types of *Symbiodinium* C. ITS2 DGGE analysis of DNA samples revealed the presence of eight *Symbiodinium* types (C1, C1b, C3, C15, C21a, C27, C30 and C40; Table 2). C15 was mainly associated with *Montipora* and *Porites*. The composition of *Symbiodinium* type D consisted of only two

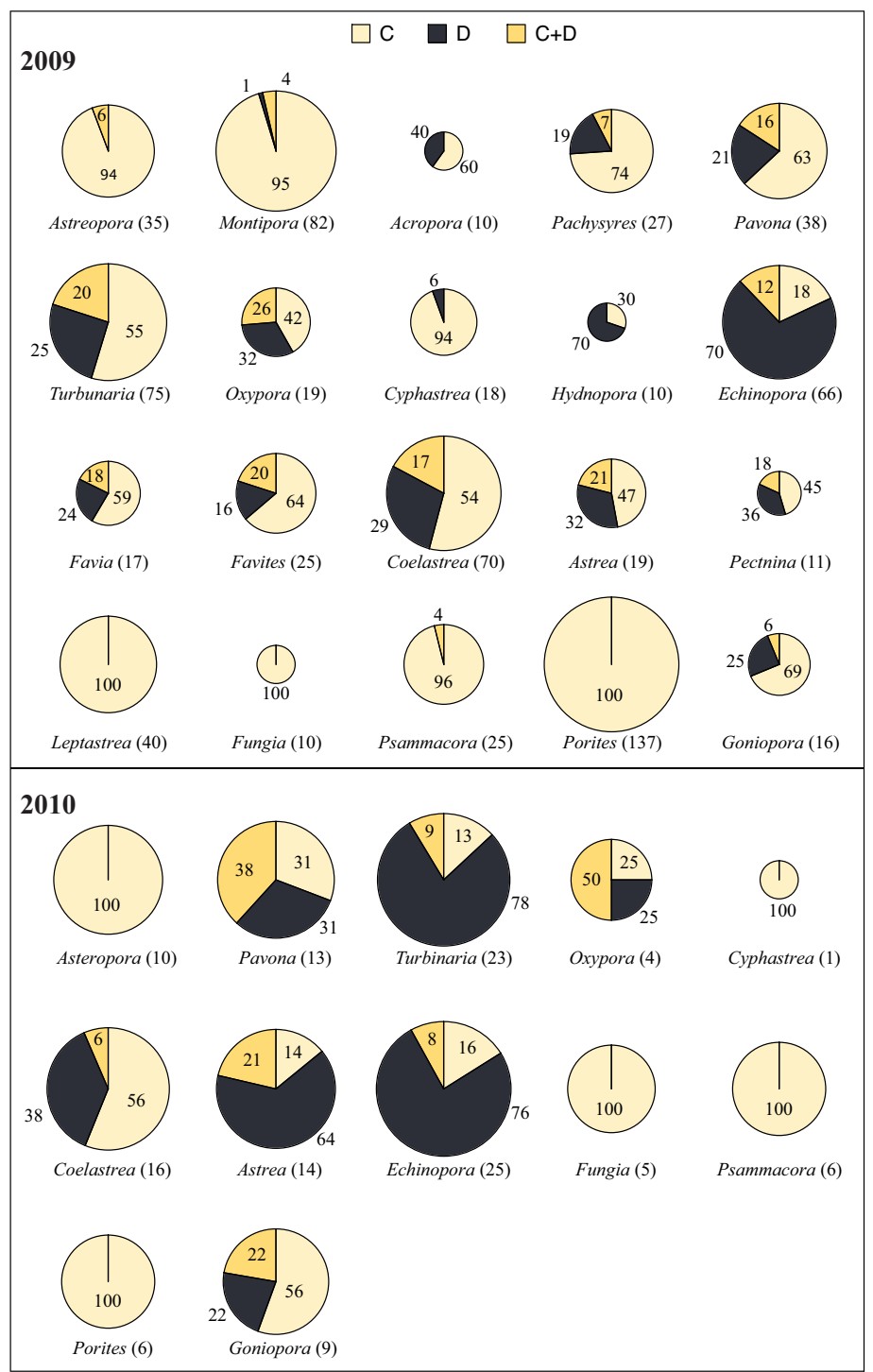

**Figure 3** *Symbiodinium* **clade distribution corals genera in 2009 and 2010.** *Symbiodinium* clade distribution (in percentages) in corals sampled in 2009 (20 genera) and 2010 (12 genera) in Dongha Atoll lagoon. The size of the pie-charts are proportional to the coral sample number. Cream, *Symbiodinium* C; Black, *Symbiodnium D*; and Yellow, *Symbiodinium* D. Sample numbers are indicated in brackets next to each genus name. Numbers inside the pie-charts indicate the percentage of clades.

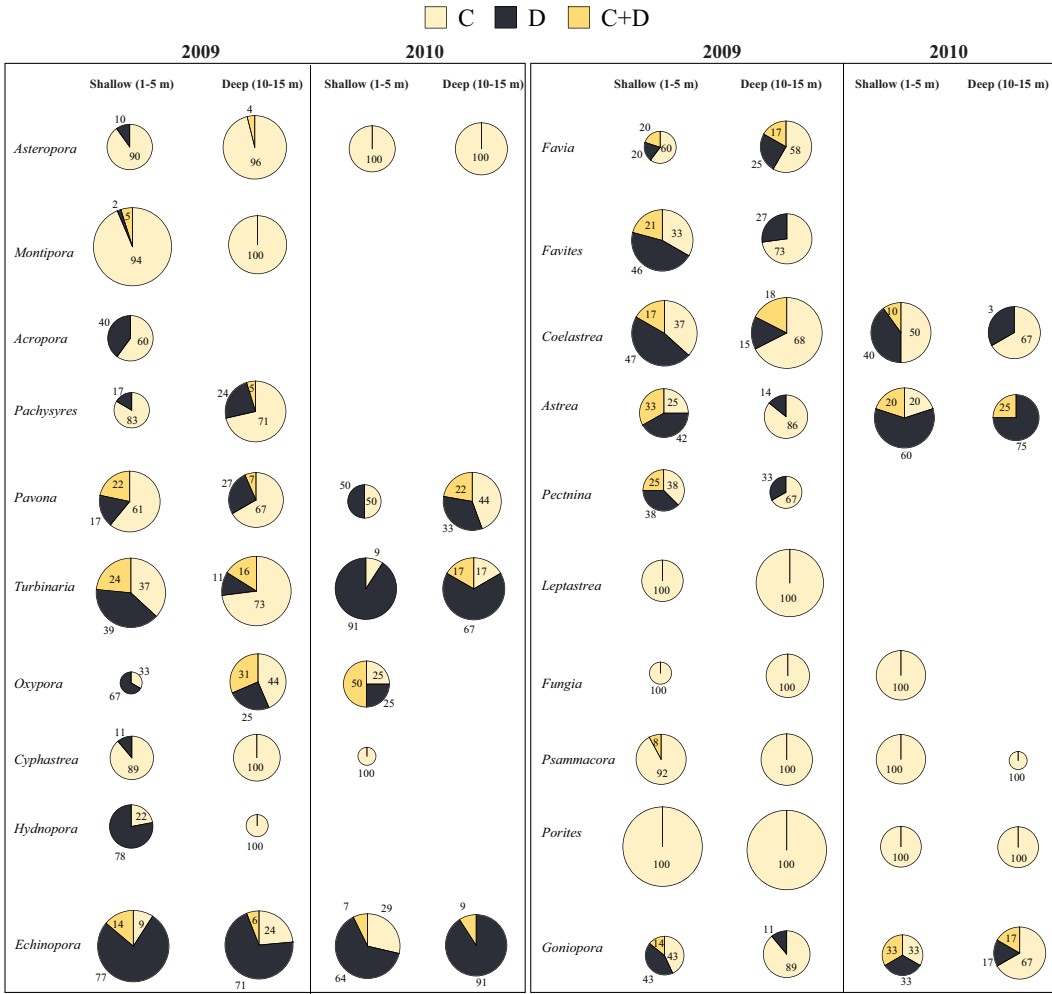

**Figure 4** *Symbiodinium* **clade distribution in corals at shallow and deep waters in 2009 and 2010.** *Symbiodinium* clade distribution in corals at shallow and deep waters in 2009 (20 genera) and 2010 (12 genera) in Dongha Atoll lagoon. The size of the pie-charts are proportional to the coral sample number. Cream, *Symbiodinium* C; Black, *Symbiodnium* D; and Yellow, *Symbiodinium* D. Numbers inside the pie-charts indicate the percentage of clades.

types; *Symbiodinium glynii* (D1) and *Symbiodinium trenchii* (D1a), either separately or in combination, depending on the coral host genus. The two species of *Symbiodinium* D are separated based on a one base-pair difference on a DGGE gel. Because of this fine difference, we ran the gels with appropriate markers for *S. glynii* and *S. trenchii* and also sequenced the cut bands from the ITS2 DGGE gels to reveal the presence of *S. glynii* in several genera of corals sampled from DAL (Fig. S1).

Out of 17 genera analyzed in 2009, the genera *Montipora, Acropora, Pavona, Oxypora, Echinopora Favites, Astrea, Coelastrea, Gonipora, Hydnopora* and *Pachyseries* were associated with *S. glynii* and/or *S. trenchii* (Table 2). Out of 12 genera analyzed in 2010, the genera *Pavona, Turbinaria, Oxypora, Echinopora, Astrea* and *Gonipora* were associated with *S. glynii* and/or *S. trenchii* (Table 2).

**Table 2  ITS2 DGGE *Symbiodinium* types in 2009 and 2010.**

| | | 2009 | | | 2010 | |
| --- | --- | --- | --- | --- | --- | --- |
| Genus | Depth | Sample number | DGGE | Depth | Sample number | DGGE |
| *Asteropora* | 3 top | 1 | C1 | 3 top | 1 | C1 |
| | 2 base | 1 | C1 | 3 base | 2 | C1 |
| *Montipora* | 2 top | 3 | C15, **D1** | | | |
| | 2 base | 1 | C15 | | | |
| | 3 top | 3 | C3, C15 | | | |
| | 3 base | 1 | C15 | | | |
| | 4 top | 2 | C15 | | | |
| | 7 top | 1 | C15 | | | |
| | 8 top | 1 | C15 | | | |
| | 10 top | 1 | C15 | | | |
| | 10 base | 5 | C15, C30 | | | |
| *Acropora* | 6 top | 4 | C1, C3, **D1, D1a** | | | |
| *Pavona* | 2 top | 1 | C1, C1b, **D1** | 3 top | 2 | C1b, **D1, D1a** |
| | 3 top | 5 | C1b + **D1, D1a**, C1 | 3 base | 1 | C1 + C1b |
| | 3 base | 2 | C1b | 6 base | 5 | C1, C1b, C27, **D1, D1a** |
| | 4 top | 3 | C1, C1b, **D1a** | | | |
| | 7 base | 1 | C1 + C1b | | | |
| | 10 base | 1 | C1b | | | |
| *Turbinaria* | 2 base | 1 | C21a | 3 top | 1 | **D1 + D1a** |
| | 4 base | 1 | C40 | 3 base | 3 | **D1 + D1a** |
| | 5 base | 1 | C1 | 6 base | 5 | C15, **D1 + D1a** |
| | 6 base | 1 | C40 | 10 top | 4 | C21a, **D1 + D1a** |
| *Oxypora* | 2 base | 1 | C27 | 6 top | 4 | C27, C15, **D1 + D1a** |
| | 10 base | 4 | **D1, D1a**, C1 | | | |
| *Echinopora* | 2 top | 2 | **D1** | 3 top | 1 | **D1** |
| | 3 top | 4 | **D1**, C40, **D1a** | 3 base | 2 | C27, **D1, D1a** |
| | 3 base | 1 | **D1a** | 6 top | 4 | C40, **D1, D1a** |
| | 4 top | 1 | C40 + **D1a** | 6 base | 1 | **D1** |
| | 5 base | 1 | **D1a** | 10 top | 1 | **D1** |
| | 10 base | 2 | C40, **D1a** | | | |
| *Favites* | 2 top | 2 | **D1** | | | |
| | 3 top | 2 | **D1, D1a** | | | |
| *Astrea* | 2 top | 1 | **D1** | 3 top | 3 | **D1 + D1a** |
| | 6 top | 2 | **D1a** | 3 base | 3 | **D1 + D1a** |
| | 8 top | 2 | C3, **D1a** | 6 top | 2 | **D1 + D1a** |
| | | | | 10 top | 1 | **D1** |
| *Leptastrea* | 6 base | 1 | C1 | 3 top | 1 | C27 |
| | 10 base | 1 | C1 | | | |

**Table 2** (*continued*)

| Genus | 2009 Depth | Sample number | DGGE | 2010 Depth | Sample number | DGGE |
|---|---|---|---|---|---|---|
| *Fungia* | 2 top | 1 | C1 + C27 | 10 top | 3 | C27 |
| | 2 base | 1 | C27 | | | |
| | 10 top | 1 | C1 | | | |
| | 10 base | 1 | C27 | | | |
| *Psammacora* | 2 top | 2 | C1, C27 | 3 base | 1 | C1 |
| | 10 base | 1 | C1 | 6 top | 2 | C27 |
| *Porites* | 2 base | 2 | C15 | 3 top | 3 | C15 |
| | 3 top | 2 | C15 | 3 base | 2 | C15 |
| | 3 base | 1 | C15 | | | |
| | 4 base | 5 | C15 | | | |
| | 10 top | 1 | C15 | | | |
| | 10 base | 1 | C15 | | | |
| *Goniopora* | 3 top | 1 | **D1** | 3 top | 1 | **D1** |
| | 3 base | 1 | C1 | 6 base | 5 | C1, **D1** |
| | 10 top | 1 | C1 + **D1** | 10 top | 2 | C1, **D1** + **D1a** |
| | 10 base | 2 | C1 | | | |
| *Coelastrea* | 3 top | 2 | C1 + **D1a** | | | |
| | 3 base | 1 | C1 | | | |
| | 8 top | 1 | C15 + **D1** | | | |
| | 9 base | 1 | C21a | | | |
| | 10 top | 1 | **D1** | | | |
| | 10 base | 1 | C1 + **D1** | | | |
| *Hydnopora* | 2 top | 1 | **D1** | | | |
| *Pachyseries* | 6 base | 1 | C27 | | | |
| | 10 base | 1 | **D1** | | | |
| | **Total** | **102** | | **Total** | **66** | |

**Notes.**
Top, Shallow; Base, Deep.

# DISCUSSION

This is the first study to investigate *Symbiodinium* association in the corals of Dongsha Atoll Lagoon (DAL). The main results of this study are that (1) *Symbiodinium* composition in DAL consists of two clades, C and D; (2) eight *Symbiodinium* C types were detected compared to only two *Symbiodinium* D types; and (3) up to 80% of coral species associated with *Symbiodinium* C underwent moderate to severe bleaching in the year 2010. Since we have utilized both RFLP and DGGE for analysis, to avoid confusion, throughout the discussion, clades and types will not be used and instead we will just use either *Symbiodinium* C or *Symbiodinium* D. We also use species name for those *Symbiodinium* types that have been designated formally.

Out of 928 samples collected from 20 (in 2009) and 12 (in 2010) genera of scleractinian corals, 598 hosted *Symbiodinium* C, suggesting its dominance in the South China Sea (see

**Table 3** Published reports of *Symbiodinium* clades diversity of scleractinian corals in genus in the West Pacific and South China Sea.

| Location | No. of genera analyzed | Clade A | Clade B | Clade C | Clade D | Clade F | Multiple clades | Reference |
|---|---|---|---|---|---|---|---|---|
| Penghu Island and Kenting, Taiwan | 26 | 0 | 0 | 26 | 2 | | 6 | *Chen et al. (2005a)* |
| Dongshan Island, China | 3 | 0 | 0 | 3 | 0 | | 0 | *Dong et al. (2008a)* |
| Luhuitou fringing reef, China | 11 | 0 | 0 | 11 | 2 | | 2 | *Dong et al. (2008b)* |
| Xisha Island, China | 25 | 0 | 0 | 25 | 3 | | 3 | *Dong et al. (2009)* |
| Zhubi coral reef of the Nansha Islands, China | 4 | 0 | 0 | 4 | 1 | | 1 | *Huang et al. (2006)* |
| Kenting, Taiwan | 16 | | | 16 | 13 | | 13 | *Keshavmurthy et al. (2014)* |
| Dong-Sha Atoll, Taiwan | 20 | 0 | 0 | 20 | 15 | | 14 | **Present study** |
| Total occurrences of clades | 106 | 1 | 0 | 106 | 36 | 1 | 39 | |

Table 3). The association of corals with *Symbiodinium* C, in addition to physiological factors, could also be due to seawater temperatures being generally below the bleaching threshold in Dongsha (Fig. 2, Table 1).

### *Symbiodinium* association in corals in 2009

In 2009, despite *Symbiodinium* C being dominant, it is interesting to highlight the differences seen in *Symbiodinium* D. There was higher proportion (24.5%) in shallow waters compared to deep waters (14.9%). Moreover, the occurrence of *S. glynii* and *S. trenchii* was higher in corals sampled from the shallow waters. The frequency of temperature fluctuations >1 °C at shallow waters was higher than at deep waters, which might be one of the reasons for the higher proportions of *Symbiodinium* D at shallow than at deep waters.

In the present study, *Astreopora*, *Pavona*, *Montipora*, *Psammocora*, *Porites*, *Fungia* and *Leptastrea* showed stable associations with *Symbiodinium* C at shallow and deep waters in DAL, suggesting that either host might be physiologically adjusted to temperature stress or their associated *Symbiodinium* C might have greater photosynthetic stability under thermal stress (*Berkelmans et al., 2004*; *Thornhill et al., 2006*; *Van Oppen, Mahiny & Done, 2005*). *Turbinaria*, *Oxypora*, *Hydnophora*, *Coelastrea*, and *Favites,* which were dominated by *Symbiodinium* D at shallow compared to deep waters, suggest flexibility in symbiont association in these corals to warmer waters via *Symbiodinium* D. The coral *Echinopora* had a stable and dominant association with *Symbiodinium* D at shallow and deep waters. In the present study, the proportion of *Symbiodinium* D showed a significant difference only in some locations, which might be due to difference distribution of coral host associated with *Symbiodinium* D.

### *Symbiodinium* association in corals in 2010 during a bleaching event

In 2010, scleractinian corals at Dongsha suffered from high seawater temperature stress caused by a La Niña event. *Astreopora*, *Cyphastrea*, *Fungia*, *Psammacora* associated with clade C including the genus *Porites*, which is generally considered as stress resistant (associated with *Symbiodinium* C15) experienced bleaching. Moreover, coral genera *Pavona*, *Coelastrea*, *Echinopora*, which were associated with clade C or C + D, also

experienced bleaching. Bleaching was seen in 84% of corals associated with clade C compared to only 10.87% of corals associated with clade D. Bleaching in those corals associated with clade D could be due to seawater temperature above threshold limits during the summer of 2010. In a previous study, the critical coral bleaching temperature threshold at Dongsha was determined to be 29.6 °C (*Dai, 2008*). During May 28 to September 6, 2010, average temperatures at shallow waters (29.9 °C) and deep waters (29.7 °C) within DAL exceeded the critical value and may have induced coral bleaching (Fig. 2, Table 1). Consistent above 30 °C seawater temperature for more than a week, both in shallow and deep waters, could have resulted in extensive bleaching in corals in 2010 (Fig. 2, Table 1). As a result of random and uneven sampling, it is not possible to analyze the correlation between *Symbiodinium* clades and seawater temperature results. We suggest that future studies in DAL should focus on obtaining samples from tagged coral colonies in different seasons and depths to understand the dynamics of coral-*Symbiodinium* associations.

### *Symbiodinium* type composition in corals in 2009 and 2010

From ITS2 DGGE analysis, DAL corals were found to be associated with eight *Symbiodinium* C (*Symbiodinium goreaui* (C1), C1b, C3, C15, C21a, C27, C30 and C40) and this number is greater than in the corals at Kenting, southern Taiwan (*Keshavmurthy et al., 2014*). Conversely, only two *Symbiodinium* D types were observed: *Symbiodinium glynii* (D1) and *Symbiodinium trenchii* (D1a). The presence of *Symbiodinium* D in corals sampled in 2010 was more obvious, and we believe this might have been due to the influence of seawater temperature (Fig. 2, Table 1). Our results also showed that *S. glynii* was dominant in many coral genera, both in 2009 (12 genera out of 20) and 2010 (seven genera out of 12), indicating that *S. glynii* may not be specific to any particular host species (*LaJeunesse, 2002*; *Baker, 2003a*; *Baker, 2003b*). This is the first work to show the presence of *S. glynii* in such a large number of coral genera and hence, we believe that *S. glynii* may not be as specific or rare, as previously thought (see *LaJeunesse et al., 2010a*).

## CONCLUSIONS

The present study has shown that *Symbiodinium* C is dominant at Dongsha Atoll in the South China Sea corals and that natural stressors, such as elevated seawater temperatures, can influence *Symbiodinium* associations in Dongsha Atoll corals. The baseline information provided on the composition of *Symbiodinium* in corals from Dongsha Atoll through the present work will help carrying out comprehensive and detailed studies on the diversity of *Symbiodinium* in corals over time. By analyzing data from normal and coral-bleaching years, we have shown that the composition of *Symbiodinium* in some corals is different between years in remote and undisturbed locations such as Dongsha Atoll. Such studies are important for understanding the future responses of corals to climate change in remote atolls. With increasing threats to coral health, mainly from temperature-induced bleaching events as seen in the Great Barrier Reef in 2016, studies documenting the *Symbiodinium* associations in corals of Donghsa Atoll will help to educate and inform park managers of Dongsha Atoll Marine National Park managers about appropriate management and conservation measures. The focus on the status of coral reefs in the South China Sea,

including Dongsha Atolls, has gained attention in the past couple of years, particularly due to the recent destruction via land-filling of several reefs several atolls in the South China Sea. Although offshore atoll reefs in the South China Sea are found to be in better condition than near-shore reefs (*Hughes, Huang & Young, 2013*), they can still undergo bleaching and mortality as a result of climate change (*Graham et al., 2006*) and other stressors (*Hughes, Huang & Young, 2013*). Trans boundary cooperation in the South China Sea region has become important for the maintenance and conservation of the various atolls and islands in the South China Sea (*McManus, 2010*). As part of the conservation program in Dongsha Atoll, it is necessary to combine the influence of natural disturbances and the effects of anthropogenic stressors, such as fishing activities, in order to understand the alterations of the present and their influences on future coral communities.

## ACKNOWLEDGEMENTS

We thank the staff of the Dongsha Atoll Marine National Park, Dongsha Atoll Research Station, Ministry of Science and Technology (MOST), Taiwan and support from members of the Coral Reef Ecology Lab, Biodiversity Research Center, Academia Sinica (BRCAS). This is Coral Reef Ecology Lab contribution No. 134.

### Funding

This work was funded by Dongsha Atoll Marine National Park (2009–2010) and the Ministry of Science and Technology (2016–2018, Grant No. 104-2621-B-001-010). The funders had no role in study design, data collection and analysis, decision to publish, or preparation of the manuscript.

### Grant Disclosures

The following grant information was disclosed by the authors:
Dongsha Atoll Marine National Park (2009–2010).
Ministry of Science and Technology (2016–2018): 104-2621-B-001-010.

### Competing Interests

The authors declare there are no competing interests.

### Author Contributions

- Shashank Keshavmurthy conceived and designed the experiments, performed the experiments, analyzed the data, wrote the paper, prepared figures and/or tables, reviewed drafts of the paper.
- Kuo-Hsun Tang and Chia-Min Hsu performed the experiments, analyzed the data, prepared figures and/or tables.
- Chai-Hsia Gan performed the experiments.
- Chao-Yang Kuo conceived and designed the experiments, performed the experiments.
- Keryea Soong contributed reagents/materials/analysis tools, logistics towards coral sampling at Dongsha.

- Hong-Nong Chou contributed reagents/materials/analysis tools.
- Chaolun Allen Chen conceived and designed the experiments, contributed reagents/materials/analysis tools, wrote the paper, reviewed drafts of the paper.

## Data Availability

The raw data has been supplied as Supplementary Files.

## Supplemental Information

Supplemental information for this article can be found online at http://dx.doi.org/10.7717/peerj.2871#supplemental-information.

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
