# Peer review of "Symbiodinium spp. associated with scleractinian corals from Dongsha Atoll (Pratas), Taiwan, in the South China Sea"

_PeerJ, doi:10.7717/peerj.2871_

## Round 0.1 · original submission · Major Revisions

The reviewers identified a number of issues regarding the organization of the paper and a lack of analysis and interpretations of the results. The authors require to address all the suggestions and comment provided by the reviewers. The authors need to carry out a major revision of the paper before it can be considered for publications.

Reviewer 1 ·

Basic reporting

In this paper, Keshavmurthy et al. examined the diversity of Symbiodinium associated with scleractinian corals from Dongsha Atoll, South China Sea. This atoll was designated marine national park since 2007. Although the scleractinian corals diversity has been studied for 30 years, this study represents the first one to provide a baseline about the Symbiodinium composition in scleractinian corals. Over 800 samples from 21 genera collected in 2009 and 132 samples from 12 genera were collected in 2010. Results showed that two clades were present (clades C and D) or a combination of both (C + D), with a dominance of clade C. Clade composition was determined using RFLPs and Symbiodinium type was determined using ITS2-DGGE. Interestingly, Keshavmurthy et al. were able to sample prior to a bleaching event (2009) and after (2010) finding an increase of corals associated with clade D. This study then, provides a basis of the Symbiodinium diversity of scleractinian corals in Dongsha Atoll.

Some comments regarding formatting were included in the attached, annotated PDF file. Nonetheless, I have some additional format comments. One recommendation that I have is spelling out numbers zero to nine and using numerals thereafter (with the corresponding exceptions). The particular cases where I would recommend spelling out numbers are when referring to the number of scleractinian families and genera sampled. Another format comment is regarding the reference to genera. In some portions of the manuscript, they are written as, for examples, “Porites spp.”, while in other sections it is referred to “genus Porites” or solely “Porites”.

My major concern is regarding figures, tables, and the text. This is because figures are not related to tables and vice versa and they are not related to what is written in the text. This can tremendously impact their results, discussion, and conclusions. Here are my specific comments:
• Alveopora is shown in Figures 3 and 4, but it is not listed as a genus sampled in 2009 (Table 2) or 2010 (Table 3). Furthermore, it is never mentioned in the text.
• Isopora is mentioned in the text as one of the 18 genera analyzed in 2009 for Symbiodinium type harboring S. glynii or S. trenchii. However, Isopora is not listed in Table 2 or shown in Figures 3 and 4.
• Astrea and Coelastrea are mentioned throughout the text as well as in Tables 2 and 3. However, they are shown neither in Figure 3 nor in Figure 4.
• Favites is mentioned in the text as one of the genera analyzed in 2010 associated with S. glynii and/or S. trenchii. Nonetheless, it is not listed in Table 3 as a sampled genus in 2010 nor there is data from 2010 shown in Figures 3 and 4.
• In Figure 4, there are more than two maroon (brown) colors, when it should only be two: one for clade C and one for clades C+D. I’m going to suggest changing either of them for another color (possibly green or blue). This will make the figure clearer for the reader. Changing the colors here also means changing the colors in Figure 3, so that they keep matching.

Experimental design

• How were the clades identified when using RFLPs? In other words, what were the positive controls or markers for clade C, clade D, and clades C+D?
• Line 177: did you use proteinase K or E?
• Lines 200-202: is the PCR protocol written correctly for the RFLP procedure?

Validity of the findings

• Lines 255-258: it is a little bit of a stretch saying that Hydnophora associated only with Symbiodinium clade C, as the sample size is one.
• It is not clear as to how the authors distinguished samples with clade C from samples with clades C+D in their results section.
• Lines 260-263: the authors stated that “at both depths, most corals were associated with more than 50% Symbiodinium clade C”, with five exceptions. From these exceptions, Oxypora, Hydnophora, and Pectinia have very low numbers (1-7 samples). Additionally, some of these exceptions are incongruent to what is shown in Figure 4. For example, although Favites has eight samples clade C and three samples clade D at the reef bases in 2009, it has 11 samples clade D at the reef top compared to eight samples clade C. A similar case is seen in Pectinia – although caution should be taken due to the small sample size. There were three samples clade D and three samples clade C at the reef top and 2 samples clade C and one clade D at the reef base. I want to reiterate that making generalizations (or exceptions) based on low sampling size is not ideal.
• Lines 260-263 cont’d: Having said this, I think the authors missed the real exception. Echinopora had the great majority of samples clade D at both depths on both years.
• Lines 260-263 cont’d: It is difficult to determine if Astrea is an exception as it was not included in Figure 4.
• Lines 263-270: I recommend adding a figure showing the sites and Symbiodinium diversity if this data is going to be explained in such detail.
• Lines 305-309: I think these sentences require revision, as they read awkward. Additionally, the authors state that they ran the DGGE gels with appropriate markers for S. glynii and S. trenchii. However, in the supplemental material figure of the ITS2-DGGE gel the marker only shows types C15 and D1. Or are Montastrea magnistellata and Turbinaria irregularis also markers? Also, it looks like one of the arrows showing D1 in M. magnistellata and T. irregularis is pointing to a heteroduplex and not to the actual band of D1. Please revise.
• Lines 305-309 cont’d: by the way, where does the marker come from? Shouldn’t it be mentioned in the Materials and Methods section?
• Lines 330-331: revise wording
• Lines 335-336: revise wording and format
• Lines 343-348: it is difficult to know if Astreopora has a stable association with clade C because this is one of the genera that has more than two maroon colors. Does it change to clades C+D in 2010 at reef tops and reef bases? This seems to be the cases as well for Montipora and Psamocora. Additionally, the taxa listed in these lines to not include Fungia, which was listed in the results section and clearly has a stable association with clade C. One final note on these lines: are the authors referring to reef tops vs. reef bases or 2009 vs. 2010 (bleaching)? The statement after “suggesting” gets confusing.
• Lines 348-350: it is not accurate to state that Turbinaria was dominated by clade D at the reef tops when 15 samples had clade D and 14 contained clade C. Importantly, the difference would be reef tops and reef bases in 2009 compared to 2010. But again, since the authors are not specifying years it is confusing to know if they are referring to the bleaching event in 2010. Additionally, Coelastrea is not in Figure 4 and its data is only included in Table 2 (2009) and not in Table 3 (2010). This makes difficult comparing proportions of each clade.
• Line 369: Figure 5 was not included in the package for reviewing. But if it were, why there is no mentioning before of Figure 5a?
• Lines 375-378: how is the possibility explained in these lines different from the first one proposed in lines 370-373?

Annotated reviews are not available for download in order to protect the identity of reviewers who chose to remain anonymous.

·

Basic reporting

Introduction and Methods need better organization in their structure. Methods section needs more clarity in explanation of sampling strategy and data analysis. Discussion section requires more in-depth synthesis of results once they are re-analyzed.
Confusing terminology: “proportion of Symbiodinium in corals” – suggests to me the ‘proportion of Symbiodinium clades C, D, C+D, within a coral ….’,
reef top & reef base terminology: suggest using ‘shallow reef & deep reef’ for more clarity
State more clearly that there was a bleaching event in 2010 (at the time sampling occurred in 2010???). This point was buried deep in Results. This is very important in interpreting your results
I am not comfortable with reporting % of clades detected. Number of samples from each genera at shallow and deep sites was not uniform so it is possible that corals sampled more extensively perhaps skewed the calculations.You are first trying to establish a baseline for Dongsha. Different genera may harbor different symbiont clades, especially depending on depth. This could be important information to report. It seems you can safely say that clade C was found in all 20 genera, both deep and shallow, during the 2009 sampling. Clade D occurred in 16(?) genera (were they deep and/or shallow?). Fourteen genera were found to contain both C & D clades (again, deep or shallow?)….
Line 42: ITS2 internal transcribed spacer 2, not internal transcribed subunit 2
Line 60: Please check ‘1.0-2.0 ºC above the summer average seawater temperatures’. As per NOAA Coral Reef Watch: One degree above the maximum monthly mean is called the "bleaching threshold" temperature. Not the mean summer SST. if this is what you are trying to say.
Line 60-61: “Bleaching results in breakdown of the symbiosis” is not really correct. Breakdown of symbiosis/Loss of symbionts results in “bleaching” …..is perhaps more correct
Line 65: This is the first time the term zooxanthelae is introduced with no explanation and is maybe thought of as an antiquated term. Perhaps stick with 'Symbiodinium' or 'symbiont' as per much of the current literature. No need to add an additional term

Line 58-69: Reorganize paragraph – it is very confusing and jumps around as you explain bleaching resistance/thermal tolerance and adaptation/acclimation/acclimatization. Be clear on the differences

Line 79: Please consider adding the pivotal references of Rowan & Powers 1991 as you are using RFLPs in your methods and their work identified the initial clades.
Rowan, R. O. B., & Powers, D. A. (1991). A molecular genetic classification of zooxanthellae and the evolution of animal-algal symbioses. Science,251(4999), 1348-1351.
Rowan, R., & Powers, D. A. (1991). Molecular genetic identification of symbiotic dinoflagellates(zooxanthellae). Marine ecology progress series. Oldendorf, 71(1), 65-73.
Line 82-83 Confusing but important sentence: ‘Among them, depending on the variation at the base-pair level, clades have been classified into “types”….’ Perhaps rephrase: ….considerable within-clade variation has been detected and classified into sub-clades (or sub-types)….
Line 88-95 Please re-organize the specificity/flexibility/shuffling explanation
Line 90 Unclear statement: “However, studies also have shown that one of the two partners is more flexible…”
Line 95 I suggest a new paragraph to explain finer scale resolution of within-clade (sub-type) tolerances and differences
Line 97 Again, ‘…Symbiodinium distribution at the type level…’ is unclear
Line 107-136 Final paragraphs of Introduction need to be re-organized to clearly understand the significance of your study area. Describing Dongsha Atoll both here and in methods is disjointed and confusing. Describing the historical uses and baseline data are important but should be in the introduction. This would be a good place to include the bleaching threshold for Dongsha, which you report much later
Describe the specifics of your various study sites in Methods.

Experimental design

Methods section needs better organization to understand how the sampling was done (experimental design). It is unclear without careful reading and re-reading to determine what samples and how many were collected each year. Very clear distinction needs to made between the bleaching status of your sampled corals each year, but especially in 2010. This has very significant impacts on how you interpret your results.
Line 156 It is unclear when the coral bleaching occurred. Both Sept 2009 and Sept 2010? It is very difficult to lump June and Sept 2009 sampling together to determine symbiont compositions of the various coral species if corals were bleached and unbleached? Or were they all unbleached at this time? Only bleached in 2010?
Line 165 HOBO pendant data logger (Onset Computer Corporation, Bourne, Massachusetts USA)
Line 171 If this protocol is described fully in your reference, then perhaps a more brief summary would be better here?

Line 187 ‘Molecular phylotyping’ First use of this term!! Please be more consistent in using terms.
I like the way you used sub-sections of methods to determine Symbiodinium clades (28S/RFLP) and then sub-clades (ITS2/DGGE). Maybe make that more clearly stated what each type of molecular method will determine by including in your sub-headings? Perhaps headings ‘Determining Symbiodinium clades’ and ‘Determining Symbiodinium sub-clades’
Line 186 what gene region for RFLP?? Large subunit 28S ?
Line 224 Temperature data analysis: This is very unclear what and how temperature data was analyzed and all extremely important. You stated temperature loggers recorded every 30 minutes. Were hourly means calculated? Daily means calculated? Were daily maxima and minima calculated and compared between shallow and deep sites and between years? Were data compared between each site (deep and shallow) and between years?
Line 227 Symbiodinium data analysis: More detail needed to understand your analysis of Symbiodinium composition. Bleached and unbleached samples all together? Each year analyzed separately or all together? I am uncomfortable with such a strategy as many studies show that symbiont composition can change between bleached and unbleached corals, also between deep and shallow. I am unfamiliar with ‘expected theoretical values’. I don’t know if Chi-Square test is valid in this situation.

Validity of the findings

To more clearly organize your methods and results I suggest using the same sub-headings when possible within each section.
Seawater data needs to be clarified so determine if analyses were appropriate.
Clarifications of comparisons of Symbiodinium clade composition between deep and shallow sites and between years (bleached and unbleached) needs to be untangled before it can be accurately assessed and conclusions drawn.
Line 234: sea water temperatures: Delete first sentence – this just adds confusion. Start with perhaps: ‘During the period June to September 2009, daily/hourly/mean(??) temperatures on the shallow reefs were….’. New paragraph: Then discuss your results from May to September 2010. It will help make the differentiate results from unbleached and bleached data more distinct. In order to analyze these data, you need to first clarify in Methods section. Is this mean hour temps? Mean daily temps?
Line 241: Table 1 Unclear what this data is. Percentage of what over 30C? hours? Days? Hours per day? 30C is used why? Why not use the local bleaching threshold, which you mention much later?
Fig 2: I would suggest making y-axis same scale 26C to 32C to more accurately compare between years; Is the y-axis seawater temperature hourly or every 30 minutes? The legend says hourly seawater temperatures but in your methods, your data loggers recorded every 30 minutes???
Fig 3: Symbiodinium composition of coral genera or Symbiodinium clades and sub-clades detected – would be more accurate. Distribution suggests to me more to spatial and temporal scales and not within genera.
Fig 4: I think the deep/shallow comparison between years would be more effective in the bar graph style as in figure 2.Or use pie charts for both Fig 2 &3 so they can be more easily compared? I really can’t visually see how does these figures change the way they are currently presented.
Is species composition different between deep and shallow sites? Perhaps this is information you can highlight and summarize from the studies done by Fang et al, Li et al, Soong et al, Dai et al)
Line 246: Table 2: Redundant information. I think Figure 2 shows the data more effectively.
Line 260: Explain why you remove Porites?
Line 276: This is the first mention of bleaching differences of your sampling, other than in the abstract. This is very important and needs to be clearly described as you open this results paragraph. It will most certainly effect how data are analyzed and interpreted.
Line 287: Symbiodinium sub-clade (ITS2/DGGE) – Bleached (2010) and non-bleached (2009) should not be grouped together. More clearly state that you are first looking at Symbiodinium sub-clades detected in each species of coral at deep and shallow sites in 2009 (unbleached). Then compare samples of the same species between years (species A/unbleached (2009) vs species A/bleached (2010))
Table 4 Needs to be re-organized and a bit more detail in the table legend so we understand clearly what data you are presenting; I would like to clearly see what sub-clade(s) were detected in each species separated by deep and shallow sites (not necessary to separate by each site) and then compare that to what sub-types were detected in that species, deep and shallow, in 2010. Perhaps also indicate which species were bleached and not bleached (differences deep vs shallow?) – this is also very important!
Line 300-306 This is really part of your discussion, not results
Line 314-317 I believe referring to these as clade C and clade D has been accepted as standard nomenclature. Using Symbiodinium C & D seems to infer that Symbiodinium C & D are a species while now it has been proposed that they may actually be genera!! Great to use species names for those sub-clades that are known.

Discussion section cannot be adequately evaluated until data is re-analyzed.
Line 358: First time I have seen the bleaching threshold mentioned for Dongsha!! This needs to be in your introduction and reflected in your data analysis and Table 1

Additional comments

I believe these data are an important contribution for the Dongsha Atoll system in the face of climate change. The introduction is rather unorganized and confusing and needs to be more carefully outlined. Some of the basic concepts are not accurately stated or explained. It seems that some of this confusion may stem from nuances in the English language and perhaps a careful review by one who speaks English as a first language will help to improve this manuscript and clarify these points.
The Methods section needs to be more clearly organized and described to understand the data collection strategy so as to properly interpret the results.
Once your methods, results, and analyses have been better defined and clarified, the discussion section needs to be re-organized and re-written. As you did not follow individual coral between years, you definitely cannot infer shuffling (line 336-337). However, you should expand on differences in symbiont composition detected between depths, between coral species and the differential thermal tolerances between the clades and sub-clade and the species composition and your bleaching observations in 2010. Discussion section will need to have more in-depth synthesis of your results and their implications for the future of Dongsha Atoll.

---

## Round 0.2 · Minor Revisions

The authors have considerably improved the manuscript by addressing the comments and suggestions provided by the reviewers in the first review. However, there are still some minor suggestions that I consider pertinent and I encourage the authors to address them.

Reviewer 1 ·

Basic reporting

Before I comment on the manuscript and edits, I want to let you know that the PDF file in the reviewed version has two versions of the abstract: the original and the modified versions. I’m not sure if this was by mistake or supposed to be like this. My other comment is that line numbers in this review correspond to the line numbers in the Word document seen in the computer. When I printed the document, line numbers changed. Additionally, I noticed that in some consecutive pages the line numbers were not consecutive. I wanted to make this clarification in order to avoid confusion to the authors as they go through my comments.

ABSTRACT
• Line 34: spell the number 7 → seven
• Line 35: spell the number 7 → seven

INTRODUCTION
• Line 90: a good reference to add about the uncertainty of the release or expelling of Symbiodinium during bleaching is Weis 2008 Journal of Experimental Biology
• Line 92: suggest changing to coral host to withstand stress, by associating with…
• Line 99: add a comma (,) before “such as”
• Lines 114-120: this is a very long sentence; I suggest breaking it down.
• Lines 121-124: suggest changing to “Symbiodinium-related stress-resistant mechanisms could be the result of a strict association with a stress-resistant clade/type of Symbiodinium or, in case of coral host with multiple Symbiodinium association, the result of shuffling between stress sensitive and stress resistant Symbiodinium clades/types”.
• Lines 129-132: suggest changing to “As coral and coral reefs face more frequent and intense bleaching events due to climate change and anthropogenic stressors, it is necessary to document the coral-Symbiodinium associations from locations that have been ignored or those that are remote”.
• Line 132: suggest changing to “A recent study from western Australia showed the presence of some unique…”
• Lines 135-136: suggest changing to “to understand the current status and predict future responses of corals in remote locations”.
• Line 191: maybe find a synonym of “untouched”.
• Line 197: spell the number 8 → eight
• Line 197: shouldn’t it be “octocorals”?
• Line 198: spell the numbers 1 and 3 → one; three
• Line 212: maybe add “Direct” before “anthropogenic disturbances”

Experimental design

MATERIALS AND METHODS
• How did you determine bleaching threshold for colonies? How many of the 132 samples in 2010 were bleached?
• Line 289: spell out 7 → seven
• Line 290: should read “seven families with 20 and 12 genera in 2009 and 2010, respectively, were collected at two depths…”
• Line 292: add a comma (,) after “four sites”

Validity of the findings

RESULTS
• Line 449: spell out 9 → nine
• Line 452: should read “locations”
• Line 453: remove italics of “clade”
• Line 457: spell out 7 → seven
• Line 491: What are the percentages of Symbiodinium C and C+D at deep waters?
• Line 495: spell out 10 → ten
• Lines 497-500: Why did you only focus on Turbinaria and Coelastrea for the difference in clade D proportions between shallow and deep water? Were these the only ones that exhibited significant differences? If so, please clarify.
• Line 503: spell out 7 → seven
• Lines 504-511: as addressed above, how did you determine bleaching?
• Line 572: add “the” between “between” and “two”
• Line 579: suggest change to “depending on the coral genus host”.
• Line 579: remove “above”
• Line 585: “out” is written twice

DISCUSSION
• Line 623: “eight” shouldn’t begin with upper case
• Lines 633-634: It could also be due to physiological requirements and/or availability of Symbiodinium for first onset of the symbiosis. In other words, temperature might be or might not be the only reason explaining the dominance of clade C – don’t rule out other options.
• Lines 637-638: This sentence is a little confusing. My suggestion: “In 2009, despite Symbiodinium C being dominant, it is interesting to highlight the differences seen in Symbiodinium D. There was a higher proportion (24.5%) in shallow waters compared to deep waters (14.9%)”.
• Line 642: This is why I think it is important to mention the percentages of C and C+D in the RESULTS section.
• Lines 645-646, 736-737 (this is one of the cases where consecutive pages did not have consecutive line numbers): why are you talking about thermal stress if in 2009 there was no bleaching event? If you’re referring to physiological plasticity at shallow and deep ranges, it is different than acclimated or adapted to thermal stress. I strongly suggest revising.
• Lines 737-739: is this because these genera are “deep” species and in DAL they can live in shallow waters? If not, this sentence sounds like this.
• Line 742: add a reference for this statement.
• Line 743: add “(data not shown)” after “locations”.
• Lines 749-752: awkward wording; please revise and remove semi-colon (;)
• Line 752: remove parentheses.
• Line 754: remove “even”
• Line 791: spell out 8 → eight
• Line 794: add colon (:) after “observed”
• Line 801: add comma (,) after “hence”
• Line 802: suggest changing to “specific or rare, as previously thought (see LaJeunesse et al. 2010a)”.

CONCLUSIONS
• Line 807: add commas (,) before “such” and after “temperatures”
• Line 809: remove “in” after “help”
• Line 845: suggest changing to “temperature-induced”
• Line 847: suggest changing to “inform”
• Line 850: suggest changing to “the recent destruction via land-filling of several atolls in the South China Sea”.
• Line 851: suggest changing to Although offshore atoll reefs in the South China Sea are…
• Line 857: add commas (,) before “such” and after “activities”.

Additional comments

Regarding the manuscript, I appreciate the effort the authors have put in responding to my comments and suggestions. My major concerns regarding figures, tables, and the text itself were successfully addressed. I have one more major concern: how did you determine bleached vs. non-bleached colonies in your sampling?

·

Basic reporting

Much improved

Experimental design

Much improved

Validity of the findings

Much improved

Additional comments

The authors have done an excellent job of incorporating comments and suggestions in this version of their manuscript. It has greatly improved the quality and clarity of their findings. I would suggest careful proof reading for a few very minor corrections.

---

## Round 0.3 · Minor Revisions

There are still few minor comments and suggestion on some of the analytical aspects of the study that need to be addressed before it is ready for publications. Please address them.

Reviewer 1 ·

Basic reporting

No comments

Experimental design

Bleaching assessment was included.

Validity of the findings

No comments

Additional comments

All major and minor comments were addressed.

·

Basic reporting

This is a very important first reporting on the Symbiodinium community at Dongsha Atoll, including observed changes during the subsequent bleaching event in 2010. Your revised manuscript shows tremendous and careful improvement over the original manuscript. I still would like more clarification in the manuscript on how temperature data was analysed. For instance, I see that you recorded temperatures every 30 minutes but were means calculated as mean hourly or mean daily temperatures? When reporting percentages >30C, the previously determined bleaching threshold for Dongsha, what was this a percentage of? Hours in each day? Or mean daily temperatures each week or each month >30C? Perhaps also consider number of days (per month or per week?) mean temps exceeded this bleaching threshold 30C? I have noted those specific lines below. All other comments are fairly minor and I have included a ‘tracked changes’ version of the manuscript with my review.

Experimental design

I still would like more clarification in the manuscript on how temperature data was analysed. For instance, I see that you recorded temperatures every 30 minutes but were means calculated as mean hourly or mean daily temperatures? When reporting percentages >30C, the previously determined bleaching threshold for Dongsha, what was this a percentage of? Hours in each day? Or mean daily temperatures each week or each month >30C? Perhaps also consider number of days (per month or per week?) mean temps exceeded this bleaching threshold 30C?

Validity of the findings

see above comments

Additional comments

November 13, 2016
Review of resubmitted manuscript
Symbiodinium spp. associated with scleractinian corals from Dongsha Atoll (Pratas), Taiwan, in the South China Sea
This is a very important first reporting on the Symbiodinium community at Dongsha Atoll, including observed changes during the subsequent bleaching event in 2010. Your revised manuscript shows tremendous and careful improvement over the original manuscript. I still would like more clarification in the manuscript on how temperature data was analysed. For instance, I see that you recorded temperatures every 30 minutes but were means calculated as mean hourly or mean daily temperatures? When reporting percentages >30C, the previously determined bleaching threshold for Dongsha, what was this a percentage of? Hours in each day? Or mean daily temperatures each week or each month >30C? Perhaps also consider number of days (per month or per week?) mean temps exceeded this bleaching threshold 30C? I have noted those specific lines below. All other comments are fairly minor and I have included a ‘tracked changes’ version of the manuscript with my review.
Line 72 In referring to a more stress-resistant Symbiodinium perhaps saying instead thermally-tolerant? The surrounding sentences refer to thermal stress so if would leave no doubt that thermal stress is the exactly what you are referring to here.
Line 149 correct form to from
Line 166 …..bleaching episode during summer of 2010 due to elevated seawater temperatures…. This might be a good place to further define the bleaching event.
Line 169 perhaps substitute ‘including’ for ‘covering’
Line 200 I suggested a minor edit to sentence structure here
Line 237 Please describe more clearly how you analyzed your temperature data. Without referring to your figures and data it is still a bit unclear. When you calculated means (± SD) were these hourly means? Daily means?
Line 240 percentage of what over 30°C? Number of days mean temperatures were above 30C? or number of hours per day >30C? Please elaborate on what data you are reporting here.
Line 253 again, what temperature data is this mean?
Line 373 I think ‘flexibiity’ is a word more commonly used to refer to corals associating with different Symbiodinium
Line 427 suggest adding ‘elevated’
Table 1. Percentage of over 30°C recorded period in 2009 and 2010 within each location.
Please also clarify here what data this table is actually showing. What is this percentage over 30C? Is it mean daily or hourly temperatures? Percentage per week? Per month? Or over how many days?
Table 2. Symbiodinium clade number in 20 genera within 7 families in 2009.
Please clarify a little more in the table legend. Is this the number of corals of each species that were found to contain each Symbiodinium clade?
Table 2. Symbiodinium clade number in 20 genera within 7 families in 2009.
Same comment
Table 4. ITS2 DGGE Symbiodinium types in 2009 and 2010
Symbiodinium should be italicized

---

## Round 0.4 · accepted · Accept

The authors have addressed satisfactorily all the comments and suggestion provided by the reviewers.